# Improvements, Variations and Biomedical Applications of the Michaelis–Arbuzov Reaction

**DOI:** 10.3390/ijms23063395

**Published:** 2022-03-21

**Authors:** Stavroula Kostoudi, Georgios Pampalakis

**Affiliations:** Department of Pharmacognosy-Pharmacology, School of Pharmacy, Aristotle University of Thessaloniki, 54124 Thessaloniki, Greece; stavroula.kostoudi@yahoo.gr

**Keywords:** Michaelis–Arbuzov reaction, phosphonates, activity-based probes, phosphonofluoridates, pharmaceutical applications

## Abstract

Compounds bearing the phosphorus–carbon (P–C) bond have important pharmacological, biochemical, and toxicological properties. Historically, the most notable reaction for the formation of the P–C bond is the Michaelis–Arbuzov reaction, first described in 1898. The classical Michaelis–Arbuzov reaction entails a reaction between an alkyl halide and a trialkyl phosphite to yield a dialkylalkylphosphonate. Nonetheless, deviations from the classical mechanisms and new modifications have appeared that allowed the expansion of the library of reactants and consequently the chemical space of the yielded products. These involve the use of Lewis acid catalysts, green methods, ultrasound, microwave, photochemically-assisted reactions, aryne-based reactions, etc. Here, a detailed presentation of the Michaelis–Arbuzov reaction and its developments and applications in the synthesis of biomedically important agents is provided. Certain examples of such applications include the development of alkylphosphonofluoridates as serine hydrolase inhibitors and activity-based probes, and the P–C containing antiviral and anticancer agents.

## 1. Introduction

The phosphonates, namely compounds bearing P–C bonds, are important molecules used in biochemistry and medicine as enzyme inhibitors (for example, FOSCAVIR^®^, the sodium salt of foscarnet (phosphonoformate), is used for the treatment of infections caused by herpes simplex virus) [1]. Certain organophosphonates used as chemical warfare agents (nerve agents) (Figure 1) have considerable toxicological properties and are potent inhibitors of acetylcholinesterase [2,3,4]. In recent years, phosphonates have been realized as an important class of activity-based probes (ABPs). Biphosphonates such as zolendronic acid are important drugs used to control calcium and manage osteoporosis and are prepared with different methods that are not based on Michaelis–Arbuzov or related-type of reactions and thus are not covered in this article but are a subject of other recent review articles [5,6]. Other organophosphate compounds are interesting industrial products used as flame retardants or in agriculture as insecticides or pesticides. Although the compounds bearing P–C bonds were not considered very abundant in nature, currently it is well-established that such substances are present in many organisms. The enzyme that catalyzes the synthesis of P–C bond in nature is called phosphoenolpyruvate (PEP) mutase and converts the PEP to 3-phosphonopyruvate. The enzyme is essential for the formation of phosphonolipids and the antibiotics fosfomycin and bialaphos (Figure 1) [7,8].

In synthetic chemistry, the most widely used and historically known reaction for the generation of P–C bond is the Michaelis–Arbuzov reaction. The reaction was first reported by Michaelis in 1898 [9] and further studied by Arbuzov in 1905 [10]. The reaction is also known as Michaelis–Arbuzov–Kaehne reaction or Arbuzov rearrangement or Arbuzov transformation or simply Arbuzov reaction [11,12]. It involves the reaction of aliphatic halides with phosphites (or phosphinites or phosphonites) to yield the respective phosphonic esters (Figure 1). During this reaction, a trivalent phosphorus is converted to pentavalent, while the alkyl group of the alkyl halide generates the P–C bond, and one of the alkyl groups from the phosphite combines with the halogen to form a new alkyl halide.

## 2. Mechanism of the Michaelis–Arbuzov (MA) Reaction

The first step in the classical MA mechanism is the attack of the lone pair of electrons of the phosphorus atom of the phosphite to the alkyl group as an S_N_2 mechanism that produces a highly unstable quasiphosphonium intermediate. This is followed by another S_N_2 reaction (Figure 2).

During the MA reaction, the newly produced alkyl halide byproduct (R_1_-X in Figure 2) acts as a reactant and competes with the original alkyl halide for reaction with the phosphite. This problem can be overcome by using a trimethyl or triethyl phosphite that generate low boiling byproducts that are removed during the reaction. Another potential way to solve this issue is to use an alkyl phosphite that will generate a less reactive alkyl halide [13]. The rate of reaction decreases with the order of RC(O)-halogen > RCH_2_-halogen > R_2_CH-halogen, while tertiary alkyl halides are unreactive. Further, the reaction is faster in the order of R-I > R-Br > R-Cl [11]. Surprisingly, the reaction of dichloroacetylene with triethyl phosphite proceeds even in cold ethereal solution to produce the monosubstituted diethyl 2-chloroethynylphosphonate at a 50% yield at 0 °C [14] and at 90% yield when conducted at −20 °C [15]. Due to strong polarization induced by the electron withdrawing phosphonate group, the 2-chloroethylphosphonate further reacts in an MA manner to yield the disubstituted product tetraethyl acetylenediphosphonate as the final product (Figure 3) [14].

In addition, cyclic phosphites can be used as reactants in the MA reaction (Figure 4) [16].

Haloalkyl phosphites are subjected to internal rearrangement upon heating, resulting in an MA product as has been observed in the case of tris-(2-chloroethyl) phosphite that rearranges to bis-(2-chloroethyl) 2-chlorethylphosphonate (Figure 5). This method is important in the production of 2-chloroethylphosphonic acid, known as Ethephon, that is generated after hydrolysis of the bis-(2-chloroethyl) 2-chlorethylphosphonate. Ethephon is an important commercial product used as a growth regulator in plants [17].

The evidence for the reaction mechanism comes from the isolation of the phosphonium intermediate under certain conditions. For example, the reaction of methyl trifluoromethanesulfonate with trimethyl phosphite results in the production of the methyltriphenoxyphosphonium trifluoromethanesulfonate as a white crystalline solid in hexane or ether, which is the stable expected intermediate for the reported double S_N_2 mechanism of the MA reaction. The stability of the intermediate is explained by the fact that the triflate is a weak nucleophile. The intermediate can further react rapidly with sodium iodide to yield the final MA product, as shown in Figure 6 [18].

A recent ultrafast 2D NMR study confirmed that the reaction between the triethyl phosphite with benzyl bromide conducted at 70 °C proceeds through the benzyl triethoxy phosphonium bromide compatible with an S_N_2 reaction. When the MA reaction is carried out with ZnBr_2_ as a Lewis acid catalyst, the second step occurs through an S_N_1 mechanism (Figure 7) [19]. The occurrence of an S_N_1 reaction is corroborated by the observation that during the MA reaction of chiral benzyl bromides with triethyl phosphite in the presence of ZnBr_2_, racemic mixtures are produced [20] (Figure 8). In conclusion, it appears that the Lewis-acid mediated MA reaction could take place through an S_N_1 mechanism.

Another exception to the regular MA mechanism is the reaction between ethyl(methyl)(2-methylhexan-2-yloxy)phosphine and methyl iodide, depicted in Figure 9, that proceeds through and the S_N_1 reaction due to stabilization of the tertiary cation attached to the phosphite [11].

Nonetheless, there are also reactions that take place through penta-coordinate phosphorous intermediates instead of phosphonium salts that are known as phosphoranes. This depends on the structure of alkyl halide or on the substituents of the phosphite. Specifically, there is evidence for phosphorane intermediates in the MA reactions of alkyl phosphites with highly fluorinated cyclobutenes [21]. The derived phosphoranes are relatively stable that decompose at room temperature to the MA product (Figure 10).

When X = F, the reaction takes place with only one substitution (Figure 11), and the resulting phosphorane **1** is stable at room temperature and slowly decomposes (but rapidly in dichloromethane) to the unexpected products shown in Figure 2.

In addition, there are MA reactions that proceed through a free-radical mechanism. Carbon tetrachloride reacts with triethyl phosphite under UV irradiation or in the presence of dibenzoyl peroxide to yield diethyl trichloromethylphosphonate through free-radical mechanism(s) [22]. Aryl halides do not generally react in a classical MA reaction since they cannot participate in S_N_2 reactions. Nevertheless, the iodobenzene reacts with phosphite through a free radical mechanism upon photolysis (Figure 12) [23].

An interesting application of the free-radical MA involves the reaction of the strained hydrocarbon [1.1.1]propellane with phosphites [24]. These reactions are called insertion reactions, since the propellane moiety is placed between the alkyl group of the phosphite and the phosphonate and are shown in Figure 13 and Figure 14.

A similar type of MA reaction occurs between the 1,3-dihydroadamantane and trialkyl phosphite, as shown in Figure 15 [25]. The adamantylphosphonic acids show promising antiparkinsonian and psychotropic activities [26].

Except aryl halides, vinyl halides are also resistant to S_N_2 reactions and thus do not react in an MA reaction. To produce vinyl phosphonates, another pathway has been described. Initially, allylic phosphonates are produced by the MA reaction that isomerize to vinyl derivatives using the tetrakis(triphenylphosphine) dihydridoruthenium (II) complex [Ru[P(C_6_H_5_)_3_]_4_H_2_] with high yields. For example, the dimethyl allylphosphonate will isomerize to the respective vinyl analogue when heated with the ruthenium catalyst for 7 h at 150 °C with a 92% yield. However, the authors did not report any preference for cis or trans isomers [27].

Taking all the above-described observations on the mechanisms that participate in the MA reaction together, it appears that these may significantly vary depending on the structure of reactants. Further, any alkyl halide except fluorides may be used as templates in MA reactions, but bromides and iodides are the most reactive and preferred reactants.

## 3. Perkow vs. MA Reactions

The reaction of trialkyl phosphites with α-halocarbonyl compounds can take place with two different pathways generating either enol (vinyl) phosphates or the classical MA products (Figure 16). The former reaction is known as the Perkow reaction [28].

The mechanism of the Perkow reaction has been investigated in detail with computational methods and is considered to proceed through the formation of an oxaphosphirane intermediate, as shown in Figure 17 [29].

A classic example of the Perkow vs. MA reaction is the reaction of 4-substituted α-bromo acetophenones with trialkyl phosphites that can take place with both mechanisms, as shown in Figure 18. The yield towards the Perkow reaction increases in the order R′ = MeO, Me, H, F, Cl, Br, NO_2_. It should be noted that the α-iodo ketones react only in an MA manner. Probably, the lower electronegativity of iodine exerts a smaller polarization effect, and further it has a greater reactivity towards displacement [11].

There is a special case of haloketone MA reaction that is observed with γ-haloketones. These react to produce 5-member ring products, as shown in Figure 19 [30].

In conclusion, Perkow reactions may act as competitive reactions for MA reactions depending on the structure of the reactants. To avoid the Perkow by-products, before performing the MA reaction, the carbonyl group should be masked with a ketal or acetal group. It should however be pointed out that the Perkow reaction is also of interest in the field of biomedical research. For example, the enol phosphates that are produced by Perkow resemble the structure of PEP (the only enol phosphate present in cells) and show phosphoryl transferring ability and can convert the adenosine monophosphate (AMP) to diphosphate (ADP) or triphosphate (ATP) [31].

## 4. Improvements in and Variations of the MA Reaction

Trimethylsilyl halides Me_3_SiX (TMSX, where X = Br, I) catalyze the rearrangement of phosphites, but also phosphinites and phosphinates to phosphonates, as shown in Figure 20. The trimethylsilyl chlorides do not react [32,33]. The mechanism of the reaction is illustrated in Figure 21 along with two representative examples. The first example in Figure 21 demonstrates the conversion of a borneol phenylphosphonate to a phosphinic derivative. Other examples with modification of natural products have been described [32].

The alkyl diarylphosphinates react in a different manner at a 2:1 molar ratio with Me_3_SiX, which is not related to the MA reaction, to yield phosphonium salts (Figure 22).

### 4.1. MA Reaction with Silyl Phosphites (Silyl–Arbuzov Reactions)

Instead of using alkyl phosphites, MA can be carried out with silyl phosphites such as tris(trimethylsilyl)phosphite. In this case, the reactivity of alkyl halides is the reverse order, RCl > RBr > RI, compared to the classic MA reaction. A further modification of this reaction involves the reaction of bis(trimethylsilyl) phosphonite (BTSP) with alkylhalide. BTSP is prepared by the reaction of hypophosphorous acid with silylating agent N,O-bis(trimethylsilyl)acetamide (Figure 23) [34]. A classic example of the silyl–Arbuzov reaction is the synthesis of lesogaberan, an investigative drug for the treatment of gastroesophageal reflux that is presented later in this review [35].

The Silyl–Arbuzov reaction has an advantage over other MA reactions when the free phosphonate acid is required. This is based on the fact that during the workup, the products simply convert to free acids with the addition of water to the reaction mixture.

### 4.2. Lewis Acid-Catalyzed MA Reaction

It was demonstrated that the Lewis acid TMSOTf or BF_3_.Et_2_O catalyzes the MA rearrangement, as shown in, for example, Figure 24 at room temperature [36].

Gollaet al. [37] developed an innovative solvent-free method to generate phenyl heterocyclic phosphinates as new antimicrobial agents against Gram positive and Gram negative bacteria. After optimization for several reaction conditions, it was concluded that the Lewis acid catalyst LaCl_3_.7H_2_O was the most efficient, leading to high yields. The reaction proceeds with heteroaryl bromides or chlorides. A representative example that exhibited the highest yield (81.5%) is shown in Figure 25.

In the same manner, phosphonites, phosphinites, and phosphites can be used in MA reactions with aryl iodides in the presence of CuI and Cs_2_CO_3_. In this modification of the MA reaction, it should be noted that the aryl bromines are unreactive [38]. Aryl phosphonates/phosphinates have also been formed at the low temperature of 40 °C using BiCl_3_ as the Lewis acid catalyst under nitrogen atmosphere [39]. Other Lewis catalysts include the Pd_2_(dba)_3_ in the presence of PhOTf and in a solvent free environment for triaryl phosphite intramolecular rearrangement [40], the CeCl_3_.7H_2_O-SiO_2_ [41], the CeCl_3_.7H_2_O [42], the already-mentioned ZnBr_2_ [19,20], the NiCl_2_ [43] that also allows the usage of aryltriflates as reactants [44], the InBr_3_ [20], the InCl_3_ that allows the reaction of N-benzyloxycarbonylaminosulfones with triethyl phosphite to generate α-amino phosphonates [45], and the NbCl_5_ [46].

Recently the use of heterogeneous BF_3_-SiO_2_ nanoparticles in a solvent-free MA reaction was presented as an eco-friendly and re-usable catalyst. It was found to exhibit superior yield when compared to other Lewis acid catalysts that were tested (e.g., ZnCl_2_, CeCl_3_). This reaction was used to produce aryl or heteroaryl phosphonates in the presence of the ionic liquid [bbmim]Br. The reaction and two examples exhibiting the highest yields of 94% are presented in Figure 26 [47]. Notably, ionic liquids that are considered “green solvents” have been found to significantly reduce the time of MA from hours to minutes as well as to reduce the temperature required to accomplish the reaction [48]. The same nanoparticles were found to operate under microwave (MW) irradiation and facilitate the reaction between urea or thiourea derivatives with triethyl phosphites to yield biologically active phosphonates with antibacterial and antifungal activity [49], as depicted in Figure 27.

### 4.3. MA Reaction with Aryl Phosphites

Aryl phosphites do not generally participate in an MA reaction due to their resistance to nucleophilic attack. Nonetheless, the addition of 5% NiCl_2_ facilitates the reaction that is carried out as shown in Figure 28 [50]. The reaction is proposed to occur through a complex mechanism depicted in Figure 29.

Without catalysts, the MA reaction of aromatic halides can take place only in cases where the aromatic ring carries electron withdrawing groups, as in the case of nitrated thienyl chlorides. These molecules are expected to have altered or improved pharmacological properties, since the thienyl group acts as a bioisostere for benzene containing molecules [51].

### 4.4. Aryne-Mediated MA Reaction

This type of MA reaction has been described for benzyne precursors such as the o-(trimethylsilyl)phenyl triflate with trialkyl phosphites and leads to aromatic carbon–phosphorus bond formation. The reaction takes place with Figure 30 in the presence of tetrabutylammonium fluoride trihydrate (TBAF.3H_2_O) at 0 °C. This method has been proposed for the preparation of a diversity-enriched chemical library that can in turn be exploited in drug screening.

Increasing the number of carbons in the alkyl substituents of the phosphite increases the yield, as demonstrated by the reaction with R = Me, yield 66%; R = –CH_2_CH=CH_2_, yield 76%; R = –CH_2_CH_2_CH_2_CH_3_, yield 93%. The MA reaction can also take place with POR(Ni-Pr)_2_ or with Ph_2_POMe with increased yields (>70%) [52]. A recent review article described in a detail a variety of reactions between organophosphates and arynes [53].

### 4.5. One Pot Room Temperature MA Reaction

A sequential reaction that involves the reaction of carbonyl halides with phosphites followed by Wolf–Kishner deoxygenation has been described. The reaction can be performed in either three steps if carbonyl halides are used or in four steps if carboxylic acids are used. In the latter case, in the first step, carboxylic acids react with oxalyl chloride to be converted in carbonyl halides [54]. The series of reactions are shown in Figure 31.

### 4.6. Ultrasound-Assisted MA Reaction

The use of ultrasound to accelerate the MA reaction has been reported with yields varying between 60 and 95% and reaction times between 1 and 2 h. Notably, when diiodomethane or dibromomethane react with trimethyl phosphite under sonication, they produce dimethyl methylphosphonate, or when they react with triethyl phosphite they yield a mixture of diethyl ethylphosphonate and diethyl haloethylphosphonate [55].

### 4.7. Electrochemical MA Reaction

This type of reaction uses electrolysis in an undivided cell with a Ni anode and graphite cathode in the presence of tetrabutylammonium chloride (Bu_4_NCl) and in hexafluoroisopropanol/DMF electrolyte under nitrogen. Importantly, it facilitates the reaction of aryl halides (mainly aryl iodides since the bromides result in significantly lower yields) with phosphites with variable yields that range between 51 and 99% for 4-bromo-1-iodobenzene and 4-trifluoromethyl-1-iodobenzene with triethyl phosphite, respectively. The reaction also takes place with methyl, isopropyl, propyl, and t-butyl phosphites but not with triphenyl phosphites [56]. The proposed mechanism is according to Figure 32 and involves the generation of an aryl radical as an intermediate. The advantage of the electrochemical methods is the fact that they can be carried out under precisely controlled conditions and can be very specific. Thus, they may be applied in the modification of complex organic molecules such as drugs.

### 4.8. Polyethylene Glycol (PEG)-Assisted MA Reaction

PEG-600 has been found to promote the MA reaction and reduce the applied temperature to 50–60 °C with yields >72% for various alkyl, aryl, or heteroaryl bromides with triethyl or trimethyl phosphite [57]. The reaction is eco-friendly since it uses mild conditions, and PEG is a non-toxic biologically compatible substance that can be recycled.

### 4.9. Solventless Continuous Flow MA Reaction

A solvent-free continuous flow MA reaction was recently reported [58]. The advantages of this method are the high purity of products and their ease of product isolation due to the lack of solvents. The experiments have been carried out in a glass microreactor chip that facilitates heat transfer under neat conditions, as in Figure 33. It works under diverse conditions and equivalents of R_2_-X. Very low equivalents of alkyl halides 0.02–0.1 are favorable for formation of homoalkylphosphonates, but equivalents 1–1.5 for heteroalkylphosphonates. With this reaction, the diethyl methylphosphonate, diisopropylmethylphosphonate, diisopropylethylphosphonate, diisopropylpropylphosphonate, and diisopropylbutylphosphonate were obtained at very high yields (up to 99%) in 8.33 min with productivity values (g/h) of 1.9, 1.6, 1.7, 1.95, and 1.8, respectively.

### 4.10. Microwave-Assisted MA

As with many other cases in organic chemistry, the microwave (MW) irradiation also facilitates the MA reaction, since it reduces the time, increases the yield, and simplifies the workup [59,60]. Further, MW irradiation facilitates the reaction between heteroaryl halides and phosphites and has been applied for the synthesis of 1,3,5-triazinylphosphonic acids (Figure 34) as new non-cytotoxic antiviral agents with, however, low to moderate activities against influenza virus H1N1 and H3N2 (the IC_50_ values were approximately 9–19-fold higher than the oseltamivir, the currently approved drug for influenza) [61].

These reactions may be carried out in a commercial microwave oven by setting the intensity to “high” inside a pressure tube, taking into consideration that heating should only take place for 1–1.5 min followed by 1–1.5 min rest and so on.

### 4.11. Alcohol-Based MA Reaction

This reaction was developed as a green chemical approach to avoid the use of alkyl halides, and it works with derivatives of benzyl or allylic alcohol in the presence of 10% tetrabutylammonium iodide (n-Bu_4_NI). It can also work with aliphatic alcohols but using triisopropyl phosphite instead of triethyl phosphite. The reaction between triethyl phosphite and alkyl alcohol yields diethyl ethylphosphonate as a byproduct and low yields of the desired product. It is possible that the I^–^ can attack both the aliphatic alkyl group and the CH_2_ of the Et–O–P (in the triethyl phosphite) to give a mixture of products. In the case of triisopropyl phosphite, the steric hindrance of the isopropyl group suppresses this reaction [62]. In addition, ZnI_2_ acts as a catalyst for the MA reaction between benzyl or allylic alcohols with triethyl phosphite [63].

### 4.12. Photochemical Michaelis–Arbuzov

The photo-MA reaction is a recent development that enables the synthesis of arylphosphonates. In this variation, a photo-active catalyst (Rhodamine 6G) is added in the reaction mixture, and the reaction is irradiated by blue light to produce aryl radicals. The reaction in carried out in nitrogen atmosphere and in the presence of diisopropylethylamine (DIPEA) as a sacrificial electron donor. Instead of trialkyl phosphites, triphenyl phosphite can also be used as a reactant. Further advantages of the photochemical reaction include the ability to be carried out under mild conditions, while it can be used for the late stage phosphonylation of bio-active molecules such as bromazepam and nicergoline (yield 14.37% and 16.66%, respectively), indicating that this “soft” method can be used to add phosphonates in complex molecules and at the same time avoid the production of byproducts (Figure 35) [64].

In the same manner, arylhydrazines react with trialkyl phosphites in the presence of eosin B (5% mole) as dye, and 1,4-diazabicyclo[2.2.2]octane(DABCO) (50% mole) as base, under visible light irradiation in the presence of air to yield aryl phosphonates with, however, moderate yields (multiple examples were given in the range of 14–74%) [65]. There is also an intramolecular photo-assisted type of MA reaction that takes place with benzyl derivatives of phosphites (Figure 36). Specifically, the reaction in Figure 36 is important for the preparation of acyclic nucleoside-based phosphonates [66].

### 4.13. S_N_Ar Mechanism in MA Reaction

The MA reaction has been applied for the synthesis of purine phosphonates, compounds that have important biomedical applications such as, for example, the inhibition of the enzyme hypoxanthine–guanine–xanthine phosphoribosyl transferase of the parasite *Plasmodium falciparum* that causes malaria. In this case, the MA reaction between halogenated purines and phosphites proceeds through an S_N_Ar mechanism that is facilitated by MW irradiation, as shown in Figure 37 [67]. An improvement of the method encompasses the use of purines that contain 1,2,3-triazoles as leaving groups that react with an S_N_Ar mechanism with phosphites. When the reaction takes place between the 2,6-bistriazolylpurines, it proceeds with regioselectivity against the 6-position (Figure 38) [68].

The above-mentioned variations of the MA reaction indicate that attempts for improvement and optimization of reaction conditions have been successfully performed. These allowed for increased yields, lower temperature, reduced reaction times, absence of solvents (or use of low toxicity recyclable solvents, i.e., PEG) that make them ecofriendly and applicable not only for laboratory synthesis but also for large scale production of organophosphonates. Additionally, soft methods such as photochemical MA have been developed that allow the conversion of complex molecules to phosphonate analogues. Otherwise, these complex molecules would have decomposed in the classical MA reaction. Further, reactants that were once thought (specifically aryl halides and triaryl phosphites) to be inert in an MA reaction have now been shown to reaction normally and efficiently using Lewis acid catalysts.

## 5. Special Cases of MA Reactions

These are MA-like reactions since they appear to be certain variations of the original reaction. The reactants are phosphites and alkyl halides, but the products are different than expected for a classical MA reaction. These differences include dehalogenation, polymerization, or altered regioselectivity, as will be outlined below.

### 5.1. Reaction of Phosphites with Vicinal Dibromides

A special case occurs between the reaction of vicinal dibromides with phosphites. When the reactant is 1,2-dibromoethane, the reaction proceeds as expected, leading to two different MA expected products depending on the molar ratio of alkyl halide and phosphite (Figure 39).

However, when there are electron withdrawing groups adjacent to both halogens in the alkyl halide, then the primary reaction is dehalogenation (Figure 40) [69].

It is interesting to note that the reaction of triethyl phosphite with 1,4-bis(trichloromethyl)benzene leads to polymerization and production of a solid polymer that is insoluble to regular solvents including sulfuric acid (Figure 41).

### 5.2. Phosphonate Derivatives of Coumarins

Coumarin moieties are found in many medicinal products, with the most widely known the drug warfarin. They also represent a potential new class of competitive HIV protease inhibitors and a new class of anti-inflammatory agents. Thus, the generation of phosphonate derivatives of coumarins may expand their medicinal properties. Triethyl phosphite reacts with coumarin halides, but the products of the reaction vary depending on the type of halogen. When iodide derivatives are used, the products are as expected for a classical MA (Figure 42), while in the case of chlorides, the reaction is more complex and yields different products (Figure 43) [70].

### 5.3. Reaction of Pyridoyl Chlorides with Triethylphosphite

The 2-, 3-, and 4- substituted pyridoyl chlorides react with triethyl phosphite, also in an unusual manner, yielding more complicated substituted products, as shown in Figure 44 [71]. Probably, the product from 2-substituted pyridoyl chloride is derived by a Perkow-like reaction of the initial aroylphosphonate, while the products from 3- and 4-substituted pyridoyl chlorides require the free acid of pyridine derivatives that is likely derived from hydrolysis of the original chloride.

### 5.4. Reactions of α, ω-Dihaloalkanes with Phosphites

The α, ω-dihaloalkanes, such as 1,2-dibromoethane, 1,3-dibromopropane, and 1,4-dibromobutane, react with trialkyl phosphites to produce the respective diphosphonates, as expected by the MA reaction [72]. Diidomethane reacts with triethyl phosphite to yield the expected disphosphonate but only with a 26% yield; thus, the MA reaction is not favorable for the synthesis of gem-diphosphonates (methylenediphosphonates) [73]. Prolonged heating of the reaction between 1,3-dibromopropane with trialkyl phosphites or attempts to purify the product by distillation yields phostones (phosphorous analogues of lactones), i.e., 2-alkoxy-2-oxo-1,3-oxaphospholanes [74] (Figure 45). In an analogous manner, the 6- and 7-membered ring phostones are derived by the reaction of phosphites with 1,4-dibromobutane and 1,5-dibromopentane, respectively [75]. Phostones can be easily polymerized through a ring opening polymerization to yield polyphosphonates. These polymers were found to be biocompatible and are expected to produce a new type of drug delivery vehicle similarly to PEG due to their “stealth” ability [74]. To produce the mono-substituted product, a large excess of α, ω-dihaloalkane should be used, or alternatively, the reaction can be performed with 1 equivalent of phosphite but with the slow addition of the phosphite in parts during the reaction [76].

This reaction is important in synthetic organophosphorus chemistry, since it can be used for the synthesis of diphosphonates; thus, the knowledge of the conditions to maximize their yields is essential. Notably, as has been described, the yield towards the disphosphonates can be efficiently controlled.

## 6. Application of the MA Reaction

### 6.1. Production of Triphenylmethyl Motif Containing Anticancer Agents

The Ph_3_C motif encompasses a new pharmacophore with important anticancer activities mediated by multiple mechanisms. Such compounds include the triphenylmethylamides clotrimazole and S-trityl-L-cysteine. To expand the diverse array of molecules carrying the Ph_3_C group, phosphonate analogues have been prepared [77,78]. As mentioned previously, the tertiary alkyl halides do not react with trialkyl phosphites. However, the phenyl containing tertiary alkyl halides react in MA, with a classic example being the reaction between triphenylmethylchloride and trimethyl phosphite that produces the expected dialkyltriphenylmethylphosphonate at 80 °C (Figure 46) [77]. Probably, the reaction proceeds through an S_N_1 mechanism, where the Ph_3_C-Cl first dissociates and then is nucleophilically attacked by the phosphorus. Support for an S_N_1 mechanism is also provided by the fact that triphenyl alcohols react with PCl_3_ to yield a P–C bond (the compound triphenylmethylphosphonyl dichloride) in an MA-type reaction. Using this reaction, the compounds shown in Figure 3 were developed as new agents exhibiting anticancer activity at the low μM range against two different melanoma cell lines [77].

### 6.2. Synthesis of Antimicrobial Agents Based on MA Reaction

#### 6.2.1. Alkylphosphonocholine Derivatives of Foscarnet

Foscarnet (Foscavir^®^) is the sodium salt of phosphonomethanoic acid and is used as an antiviral agent for the treatment of infections caused by herpes viruses (cytomegalovirus, HSV1 and HSV2). Foscarnet resembles the structure of pyrophosphate and inhibits the activity of the viral DNA polymerases. It is produced by the MA reaction between triethyl phosphite and ethyl chloroformate followed by hydrolysis of the derived phosphonate ester with NaOH (Figure 47). Other similar esters of phosphonomethanoic acid have been synthesized and tested for their anti-herpes activity, and some were found to have the same effectiveness with the parent compound in inhibiting HSV1 replication in vitro [79].

In addition, alkylphosphocholines (APCs) derivatives with the foscarnet moiety have been synthesized for potential cytotoxic and antimicrobial activity with the set of reactions shown in Figure 48 [80]. Compound **3** was the most cytotoxic agent against the human cervical adenocarcinoma cell line (HeLa) and human breast adenocarcinoma cells (MCF-7), whereas compounds **2** and **1** exhibited the highest anticandidal and antiprotozoal activity, respectively.

#### 6.2.2. Methylenecyclopropane Phosphonate Derivatives as Antivirals

The synthesis of these compounds is carried out according to Figure 49 [81]. The importance of this study is based on the generation of the methylenecyclopropylphosphonate backbone that can also be used in other biomedical applications.

#### 6.2.3. Phosphonate Derivatives of N-Phenylsulfonamide with Antimicrobial Activity

These compounds were synthesized using the MA reaction according to Figure 50. Among the compounds, compound **13** was the most active against bacterial and fungal strains as determined by the minimum inhibitory concentrations (MIC) against the bacteria *S. aureus* (15 μg/mL), *B. subtilis* (20 μg/mL), *E. coli* (25 μg/mL), and *K. pneumoniae* (15 μg/mL), and the fungi *C. lunata* (15 μg/mL) and *A. niger* (20 μg/mL) [82].

#### 6.2.4. Phosphonate Antiviral Prodrugs

A representative example is a class of phosphonate prodrugs of azidothymidine (AZT), an antiviral nucleoside analogue with anti-HIV activity. These prodrugs mimic a phospholipid; thus, they have increased lipophilicity and can interact with plasma membranes. The phosphonate derivative of AZT has been obtained by the MA reaction between the appropriate alkoxyalkyl bromide with trimethylphosphite and subsequent hydrolysis of the phosphonate ester with TMS bromide, as shown in Figure 51. The coupling with AZT was carried out with DCC [83].

### 6.3. Phosphonate Surfmers

Surfmers (a combination of words surfactant and monomer) are a class of surfactants. Phosphonate surfmers have been designed that can participate in polymerization reactions and thus constitute an integral part of the polymer. In the surfmer, the phosphonate group consists of the hydrophilic head, while the hydrophobic tail ends with a methacrylate group that is amenable to polymerization in a microemulsion to generate the surfmer nanoparticles. The synthesis of the surfmers is shown in Figure 52. The nanoparticles exhibit increased cellular uptake without exhibiting cytotoxicity; thus, they can be used for drug delivery. In addition, they act as platforms for biomimetic mineralization of hydroxyapatite [84].

### 6.4. Phosphonate Dendrons

Phosphonate-based dendrons have been developed as biocompatible coatings for superparamagnetic iron oxide (SPIO) nanoparticles for magnetic resonance imaging (MRI) combined with therapy by hyperthermia in a theranostic manner. The development of the phosphonate dendrons was based on their high affinity for iron oxides. Multiple phosphonate anchors were developed as coatings, and two examples of mono or bis-organophosphonate anchors that were developed using the MA reaction are shown in Figure 53. Then, the polyethylene glycol or the poly(amido)amine (PAMAM) units were attached to the anchor [85].

### 6.5. Development of Haptens

The production of monoclonal antibodies to identify the aged and non-aged OP–acetylcholinesterase conjugates is of major importance in toxicological research. To accomplish this task, the antibodies should recognize the monosubstituted phosphonylated active site serine (aged) compared to the disubstituted (non-aged). To produce such specific antibodies, two specific haptens have been developed: the diethoxyphosphoryl (mimicking the non-aged); and the monoethoxyphosphoryl (mimicking the aged). These haptens were attached on the keyhole limpet hemocyanin (KLH) or bovine serum albumin (BSA) carrier for immunization using EDC/NHS coupling chemistry. When the KLH conjugates were injected in mice, they triggered the production of antibodies and using hybridoma technology, monoclonal antibodies were selected. The use of these specific antibodies will allow the toxicity of organophosphorus compounds to be assessed and monitored (especially of paraoxon) in animal models. The complete synthesis of haptens is shown in Figure 54 [86].

It should be noted that there are alternative methods to develop specific haptens for organophosphorus agents that include the reaction between the organophosphate compound with peptides derived from the protein of interest (e.g., AChE) and that encompass the reactive residue [87,88]. However, the difference between the previously described methodology and these alternative methods is the fact that by using the former method, it is possible to produce well characterized mimics of aged enzymes, while the reaction of orgaphosphates with peptides could possibly lead to development of “mixed” or only “non-aged” compounds.

### 6.6. Other Phosphono-Derivatives with Potential Biological Activities

#### 6.6.1. Phosphonylated Sugar Derivatives

MW irradiation has been applied for the MA mediated synthesis of phosphonate derivatives of sugars that constitute intermediate compounds towards the synthesis of phosphonate nucleosides with biomedical applications. The reaction was carried out as shown in Figure 55 for 90 min at 220 °C and at 300 W power of MW irradiation [89].

#### 6.6.2. Polyphosphate Analogues

Polyphosphates are found in many biochemically important molecules including the triphosphate nucleotides, in co-enzymes, and in isoprenoids (namely isopentenyl pyrophosphate, dimethylallyl pyrophosphate). The design and synthesis of non-hydrolysable stable analogues of polyphosphates that act as enzyme inhibitors is of great importance in biomedical research to probe their functions. One way to synthesize stable analogues of triphosphates is to connect the phosphorus atoms with methylene bridges. To construct these molecules, the MA reaction between triethyl phosphite and ethyl bischloromethylphosphinate was conducted under reduced pressure, as shown in Figure 56, with an 85% yield [90].

However, the final symmetric phosphonate product could not be easily attached to nucleosides. Thus, the reaction shown in Figure 57 was chosen as an alternative that generates asymmetric phosphonates that can easily be attached to nucleosides by the Mitsunobu reaction [90].

#### 6.6.3. Autophagy Simulators

The 2-substituted-3-phosphon-1-thia-4-aza-2-cyclohexene-5-carboxylates have been synthesized as analogues of the natural metabolite lanthionine ketimine (LK). The LK and its ethyl ester (LKE), which is more cell permeable, have important neuroprotective functions that are mediated by stimulation of autophagy. Importantly, the LKE was shown to increase the life span of the amyotrophic lateral sclerosis Tg-*SOD1^G93A^* mouse model [91] and slow the decline of cognitive symptoms in the 3xFAD mouse model of Alzheimer’s disease [92]. The reported phosphonate analogues (Figure 58) were synthesized to investigate whether they have increased biological potency. Indeed, it was found that the 2-isopropyl-LK-P was a more potent autophagy stimulator than the LK at 10 μM. The synthesis was started from the MA reaction between the appropriate acyl chloride and triethyl phosphite, as shown in Figure 58 [93].

#### 6.6.4. Lesogaberan

Lesogaberan is a drug under study for the treatment of gastroesophageal reflux. The synthesis of lesogaberan is shown in Figure 59 and requires the modified MA reaction between (R)-2-fluoro-3-iodopropylamine and BTSP that was described previously [35].

#### 6.6.5. Development of α-Diaryl Substituted Phosphonate Compounds

The α-diaryl substituted phosphonates have important biomedical applications including antioxidant and antitumor activities [94]. Recently, the MA reaction between para-quinone methides, which are electron-deficient alkenes, with alkyl phosphite or aryl phosphonite or diarylphophinite was accomplished in the presence of silver tetrafluoroborate (AgBF_4_) as catalyst. This modified MA reaction does not use alkyl or aryl halides and allows the preparation a variety of α-diaryl substituted phosphonates [95]. The reaction proceeds as shown in Figure 60, and the yields are between 75 and 96% for a variety of substituents.

#### 6.6.6. Surrogates for V-Type Nerve Agents

The O-4-nitrophenyl O-ethyl methylphosphonate, a surrogate for methylphosphonothioate VX (and other V-agents such as VM, etc.) has been synthesized (Figure 61) to assist in the development of new antidotes and medical treatments or to assess new detection methods and test protective equipment [96]. This surrogate still inhibits AChE, but to a significantly lower extent, and it is thus much safer to use for experiments.

## 7. Alkylphosphonofluoridates

These are compounds with the general formula RP(O)(OR′)F. When the R ≤ 3C (methyl, ethyl, propyl, and isopropyl), these compounds are potent irreversible inhibitors of acetylcholinesterase (AChE) and constitute the G series of nerve agents that are covered by the Chemical Weapons Convention (CWC). On the other hand, alkylphosphonofluoridates with R ≥ 4 display significantly reduced toxicity and are interesting as other than AChE enzyme inhibitors or as activity-based probes (ABPs). Below, certain examples of MA mediated synthesis and applications of alkylphosphonofluoridates are given.

### 7.1. Synthesis of Radiolabeled (C14) Sarin

Sarin is a highly toxic organophosphate compound used as nerve agent. Like all nerve agents it acts as a potent irreversible inhibitor of AChE. Nonetheless, it has other biological functions and targets as well. To monitor the action of sarin in vivo in experimental animals, it is necessary to have a radiolabeled compound. The other option is to introduce non-radioactive tracers such as a fluorescent group, but these will significantly alter the toxicokinetics of the parent molecule. To produce a nerve agent for biomedical applications, the MA reaction is the most classic one. Specifically, radiolabeled sarin is produced from readily available radiolabeled methyliodide and triethylphosphite, as shown in Figure 62, with an overall yield 61% [97].

### 7.2. Cyclohexyl Alkyl (or Aryl) Phosphonofluoridates

These compounds are potent inhibitors of chymotrypsin and chymotrypsin-like enzymes. Their structure resembles the nerve agent GF (cyclohexyl ester of methylphosphonofluoridic acid); however, a bulky aromatic group is attached to the phosphorus atom that enhances the specificity for chymotrypsin and related enzymes and significantly reduces the toxicity. The most potent chymotrypsin-like inhibitors are the ones shown in Figure 4 [98].

### 7.3. Alkylphosphonofluoridates as Activity-Based Probes (ABPs)

Activity-based probes (ABPs) are small molecules that covalently bind to the active enzymes and enable their detection either in vitro or in vivo. ABPs are per se suicide inhibitors. These significant tools consist of three parts: the reactive group (or warhead), a spacer (that may include an enzyme recognition sequence if increased specificity is required), and a reporter tag that can be an affinity tag or a fluorophore. The initial work aimed to design a phosphonofluoridate ABP against serine hydrolases [99]. This ABP was a biotin (or fluorescein)-tagged alkylphosphonofluoridate and was named FP-biotin (or FP-fluorescein). The FP-biotin was synthesized based on the reaction of 11-iodoundec-1-ene with triethyl phosphite. The next step was partial hydrolysis by TMS bromide, followed by oxidation with RuCl_3_/NaIO_4_ to the respective acid, reaction with diethylaminosulfur trifluoride (DAST), and finally an N-hydroxylsuccinimide (NHS)-assisted attachment of biotin or fluorescein tag (only the biotin tag is shown in Figure 63).

To map the proteolytic activities in a biological sample, the FP-biotin is allowed to react with the sample (e.g., the cellular extract), and then the sample is run on SDS-PAGE, and the FP-reacted bands are identified with Western blot using a streptavidin–HRP-based detection system.

In the same direction, another biotinylated phosphonofluoridate ABP named B24P (Figure 5) has been synthesized. This ABP was used to design a new histochemical method to spatially map the enzymatic activities in situ in tissue cryosections that was called activography [100]. Application of activography with B24P in skin cryosections from patients suffering from various skin diseases that differ in their genetic background showed that the degree of epidermal desquamation correlated with the epidermal enzymatic activities, namely the higher the desquamation and the stronger the epidermal proteolytic activities [101].

### 7.4. Click Chemistry and ABPs

In this case the ABP is separated into two molecules that are linked together with click chemistry. This ABP is designed to consist of the fluorophosphonate reactive group (warhead), a spacer, and a 1-alkyne moiety that is amenable for Cu(I)-catalyzed coupling with azides connected with a detection tag. The classical reported example is the synthesis of the FP-alkyne (Figure 64) [102]. The FP-alkyne is used as described with the FP-biotin, namely it is allowed to react with cellular lysates or any other biological/clinical specimen under study, and the active proteases are 1-alkyne modified. In addition, the FP-alkyne can be used to label intracellular proteases in cell culture. Then, the cells are harvested, lysed, and incubated with the azide-modified fluorescein tag to perform the Cu(I)-catalyzed [3+2] cycloaddition (Huisgen reaction) and label the active proteases (Figure 65).

The advantage of using FP-alkyne compared to other ABPs that carry the detection tag is that the bulky reporter group (detection tag) can significantly reduce cellular permeability, thus reducing the ability of the molecule to reach intracellular enzymes in vivo or in cellsin vitro. Further, it may limit the accessibility to the active site of the enzyme, although in many cases this issue can be dealt by appropriately adjusting the length of the spacer. Indeed, it was demonstrated that the FP-alkyne crosses the cellular membranes of Caco-2 cells as expected, while the bulky FP-fluorescein did not. The specificity of the FP-alkyne to label only the active form of the proteases was validated with active KLK7. FP-alkyne was able to react only with active KLK7, not with the zymogen or with KLK7 preincubated with a reversible inhibitor [102]. The ABP where the azide group instead of the 1-alkyne and has been attached to the fluorophosphonate warhead through a hydrocarbon spacer has also been designed. In this case, the detection was carried out with an alkyne-modified tag [103].

## 8. Phosponates as Intermediates in the Synthesis of Alkenes

An important reaction in synthetic chemistry is the coupling between phosphonates and aldehydes that results in the production of trans-alkenes. The MA reaction is used for the preparation of phosphonates. This reaction has been applied for the synthesis of stilbenes (and the heterocyclic analogues 2-stilbazole, 2-styrylfuran, and 2-styrylthiophene) using diethyl benzylphosphonate and the corresponding aldehyde with sodium methoxide in DMF with yields of 75–85% (Figure 66) [104]. Certain stilbenes that contain fluorine atoms, such as the compound shown in Figure 6, have been prepared in an analogous manner and found to exert potent anticancer activity both in vitro and in vivo in nude mice xenotransplanted with LS174 colorectal cancer cells [105]. In another study, such fluorine-containing stilbenes were found to have potent anti-NF-kB activity, while they could upregulate the activity of the Nrf2 transcription factor that controls the antioxidation potential of the organism. Thus, these compounds are expected to have potent anti-inflammatory action.

In the same manner, the MA reaction has been applied for the synthesis of the phosphonate intermediate **18** used for the preparation of β-carotene with sodium methoxide in pyridine at 0 °C (Figure 67) [106].

Using the same methodology, the compound nostodione A (Figure 68) was synthesized [107]. Nostodione A is an alkaloid found in cyanobacteria with antimitotic and proteasomal inhibition properties. In addition, nostodione A is a highly UV-absorbing agent that can be used as an ingredient in sunblock products. Nostodione A, and more importantly the compound with the same chemical scaffold shown in Figure 7, have potent anti-*Toxoplasma* activity.

The phosphonate intermediates have allowed for the easy preparation of a large variety of substituted alkenes that show important and diverse biomedical applications. Especially for the synthesis of stilbenes, which show potent antioxidant and anti-inflammatory actions, this method is the most preferred.

## 9. Conclusions

The organophosphates encompass a major class of biomedically and industrially important materials. Thus, significant progress in the reactions that generate phosphorous-carbon bonds has been made [5,35,108,109,110]. Nevertheless, this rapid expansion in organophosphorus chemistry has called for the detailed review and discussion of the progress that has been made in certain named reactions that generate P–C bonds. The MA reaction is the most classic and a reaction of great importance for the generation of organophosphates, allowing for the incorporation of a broad range of reactants. Here, a detailed presentation of the improvements and variations of the MA reaction is given, with potential applications in the biomedical field. Previous review articles have not dealt in detail with the MA reaction and its applications since 1985 [5,11,35,108,109,110].

The variations of the MA reaction have made possible the use of aryl halides or triaryl phosphites as reactants, while they have enabled light to be shed into its mechanism. Interestingly, there is no universal mechanism that can explain all the MA reactions, but the participating mechanism differs according to the reactants and/or the presence of a catalyst. In addition, an extended discussion in the context of biomedical applications was given to demonstrate the applicability of this relatively straightforward reaction in the production of new pharmaceutical/biomedical agents. Indeed, the organophosphonates synthesized by the MA and the MA-related reactions are potential drug candidates due to their anticancer, antiviral, antiprotozoal, and antibacterial activities.

## Data Availability

Not applicable.

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
