# Peer review of "Improvements, Variations and Biomedical Applications of the Michaelis–Arbuzov Reaction"

_ijms, 2022, doi:10.3390/ijms23063395_

Round 1

Reviewer 1 Report

If the focus of this review is the "Biomedical Applications of MA reaction", the entire review should be classified basing on the Bio-applications, and not on the reaction types. In fact, literature examples cited in this review are not always related with the Title of the manuscript.

The real section tuned with the title of the manuscript seems to be the 6. Application of the MA reaction. The previous pages are not consistent with the target of the review. Thus, in my opinion, the authors need to choose if remove the other pages, or change the Journal, considering the submission in a more specialized Organic journal.

Some other issues:

In the Introduction section, references about FOSCAVIR and Nerve Agents are required.

Scheme 1 and 2 show different substrates. In the MA reaction, the substituents of P contain the oxygen atoms?

it is not clear for me the connection of the reactions described in the section 2 with the general mechanism of reaction

In Scheme 7, oxygen atoms of the P-Zn intermediate missing

Author Response

Reviewer 1

If the focus of this review is the "Biomedical Applications of MA reaction", the entire review should be classified basing on the Bio-applications, and not on the reaction types. In fact, literature examples cited in this review are not always related with the Title of the manuscript.

The real section tuned with the title of the manuscript seems to be the 6. Application of the MA reaction. The previous pages are not consistent with the target of the review. Thus, in my opinion, the authors need to choose if remove the other pages, or change the Journal, considering the submission in a more specialized Organic journal.

Answer:

We have changed the title to “Improvements, variations and biomedical applications of the Michaelis-Arbuzov reaction”. The title now successfully depicts all the information that is included in the text.

We would like to note that understanding the new developments in MA reaction is important to design new synthetic procedures (parts 3, 4 and 5). Thus, we believe that the first part of this paper is important and has not been a subject of a recent review article in detail. Further, within this part certain examples that find biomedical applications can be found (e.g. phosphonate derivatives of coumarins). On the other hand, part 2 depicts the mechanism of the reaction that is also necessary to understand the MA reaction and how we can expand the library of reactants. Finally, parts 6-8 are all the bioapplications of MA.

Some other issues:

In the Introduction section, references about FOSCAVIR and Nerve Agents are required.

Answer

New references (1-5) have been added as suggested.

Scheme 1 and 2 show different substrates. In the MA reaction, the substituents of P contain the oxygen atoms?

Answer

Both schemes show the same substrates when R=alkoxy group in Scheme 1. However, to make this clear we have now changed Scheme 2 in accordance to Scheme 1.

it is not clear for me the connection of the reactions described in the section 2 with the general mechanism of reaction

Answer

The reactions depicted in Section 2 demonstrate the variations in the MA reaction. Although, the classical MA reaction entails a mechanism of two SN2 reactions, this may be significantly vary depending on the structure of the alkyl or aryl halide and the conditions. To highlight this, we have included all the potential mechanisms that have identified in MA reaction in Section 2. For example, the general MA mechanism cannot account for the reaction with aryl halides or for the Lewis acid mediated catalysis. In addition, these reactions set the basis to expand the library of reactants that can participate in the MA reaction. Thus their discussion is essential for this review article.

In Scheme 7, oxygen atoms of the P-Zn intermediate missing

Answer

The Scheme has been revised accordingly.

Reviewer 2 Report

C-P bonds play an important role in organic and medicine chemistry. Great achievement has been obtained in the area of C-P bond formation. Pampalakis and coworkers described the developments andapplications of theMichaelis-Arbuzov reaction.  Before this manuscript was published, some issues should be addressed.

  1. The position of the picture in the text needs to be adjusted and is not aligned.
  2. The reaction equation is ugly and needs to be modified.
  3. There should be a general conclusion and summary behind each part, not just a simple description of the work of the literature.

Author Response

Reviewer 2

C-P bonds play an important role in organic and medicine chemistry. Great achievement has been obtained in the area of C-P bond formation. Pampalakis and coworkers described the developments andapplications of theMichaelis-Arbuzov reaction.  Before this manuscript was published, some issues should be addressed.

  1. The position of the picture in the text needs to be adjusted and is not aligned.

Answer

All pictures have been aligned in the text. Scheme 29 and Figure 7 are the only pictures that extend from the text limits but they are still in accordance with the journal style and requirements.

  1. The reaction equation is ugly and needs to be modified.

Answer

All figures have been re-adjusted. Certain figures (Figure 1, 3 and Scheme 48) have been designed from the beginning exclusively in ChemDraw to improve quality.

  1. There should be a general conclusion and summary behind each part, not just a simple description of the work of the literature.

Answer

A general conclusion has been added in each part as suggested by the Reviewer.

Round 2

Reviewer 1 Report

the authors modified the manuscript is some parts, however, my idea that this review should be more appropriate in an organic journal is unaltered.

Author Response

We have highlighted the biomedical applications in the parts 2-5 of our manuscript. Specifically, these have been highlighted in section 2 (lines 84-87 and 155-156), section 3 (lines 198-203), section 4 (lines 209-212, 279-280, 294-296, 301-303, 335-338, 404-405, 420-421) and section 5 (494-497). Certain examples of biomedical applications of the MA reaction in these sections e.g. coumarins (section 5.2) and lesogaberan and phenyl heterocyclic phosphinates (section 4.1), were already shown in R1 draft. These examples together with the new additions provide a more detailed connection between the MA and the biomedical applications. It should be noted that parts 6-8 already demonstrate the biomedical applications of the MA reaction.

Round 3

Reviewer 1 Report

I appreciate the efforts done to modify the manuscript. It can be now published